# Phytogenics and encapsulated sodium butyrate can replace antibiotics as growth promoters for lightly weaned piglets

**Caio Abércio da Silva**[1]*, **Cleandro Pazinato Dias**[2‡], **Marco Aurélio Callegari**[2‡], **Gabrieli de Souza Romano**[2‡], **Kelly Lais de Souza**[2‡], **David Vanni Jacob**[3], **Alexandre José Ulbrich**[3], **Tim Goossens**[3]

1 Animal Sciences Department, Center of Agrarian Sciences, State University of Londrina, Londrina, Paraná, Brazil, 2 Akei Animal Research, Fartura, São Paulo, Brazil, 3 Nutriad Animal Nutrition Ltda., Campinas, São Paulo, Brazil

☯ These authors contributed equally to this work.
‡ CPD, MAC, GSR and KLS also contributed equally to this work.
* casilva@uel.br

**Data Availability Statement:** All relevant data are within the paper and its Supporting information files.

## Abstract

The objective of this study was to evaluate the effect of essential oils plus dry herbs (PHYTO) and encapsulated sodium butyrate (BUT) supplementation compared with enramycin (ENR), as a growth promoter, on the performance, diarrhoea control and intestinal microbiota in lightly weaned piglets. Two hundred weaned piglets, 20 days old, 4.69 ± 0.56 kg, were submitted during the nursery phase (20 to 69 days of age) to four treatments: control (CTR)—without any additive supplementation; ENR (with 8 ppm of enramycin throughout), BUT (with 2000 ppm between 20 to 34 d, 1500 ppm between 34 to 48 d and 1000 ppm between 48 to 69 d), and PHYTO (150 ppm between 20 to 48 d). At 62 days old, forty piglets (10 replicates per treatment) were slaughtered to perform bacterial identification through 16S rRNA (V3-V4) sequencing of the caecal content. During the second phase of the trial (34 to 48 days), the BUT group showed higher DWG (P = 0.023) and BW (P = 0.039) than the CTR group, and all groups that received additives had better FCR than the CTR group (P = 0.001). In the last phase of the trial (48 to 69 days), the ENR group presented a better FCR (P = 0.054) than the CRT and other groups. In the total period (20 to 69 days), ENR and BUT showed better FCR (P = 0.006) than CRT. Diarrhoea incident data showed differences (P<0.05), favouring the BUT treatment compared to the CTR. Only the *Megasphaeraceae* and *Streptococcaceae* families showed differences (p<0.05) in relative abundance between CTR and PHYTO and between CTR and BUT, respectively. Differential abundances of the *Megasphaera* and *Streptococcus* genera were observed between CTR and PHYTO and CTR and BUT. Phytogenics and encapsulated sodium butyrate are able and effective for modulating the specific caecal microbiota, improving performance and controlling diarrhoea occurrence.

**Funding:** The authors received no specific funding for this work.

**Competing interests:** The authors have declared that no competing interests exist.

## Introduction

Subtherapeutic use of antibiotics in livestock feed has been known to improve animal health, growth, and feed efficiency as well as the quality of the end products intended for human consumption [1, 2]. Despite these beneficial effects, the current worldview calls for a ban on the use of antibiotics as growth promoters, as they may lead to the development of antibiotic resistance in humans through their consumption in food [3, 4], although a full consensus has not yet been reached [5].

Restricting the use of these molecules is particularly detrimental for weaned piglets, as they face numerous stressors during this critical phase [6]. Furthermore, with the increase in sows' prolificacy, piglets at birth are lighter, presenting poorly developed digestive systems [7], being more susceptible to infectious intestinal disease and energy deficiency [8], compromising their health at weaning. In addition, weaned piglets have high digestive enzyme insufficiencies [9] and limited ability to secrete HCl, which can lead to poor nutrient absorption [10] and consequently to increased osmolarity in the digestive tract and potentially to osmotic diarrhoea, ultimately resulting in lower growth performance [11].

Given the lack of evidence and growing concerns, identifying and developing applicable alternatives to in-feed growth promoting antibiotics has become a priority [12]. In this sense, phytogenics and butyrate are good examples of potential nonantibiotic substitutes. Butyrate is the salt of an acid (butyric acid) that regulates gene expression, cell differentiation, immune modulation, oxidative stress reduction, and diarrhoea control [13]. It also provides energy for colon mucosa cells and helps promote the development of gastrointestinal mucosa, improving pig health and performance [14–16]. When this acidifier is supplemented in a protected form, such as in a fat matrix, it is released slower and is able to reach the lower gastrointestinal tract, where it is most beneficial [13].

Phytogenic can act as digestive stimulants [17]. They are able to modulate the intestinal microbiota [18], stimulate endogenous secretion [19] and improve food palatability, thus increasing feed intake and weight gain [4, 20]. This study considered the challenges of weaning and the restrictions on the use of antibiotics as growth promoters and aimed to evaluate potential alternatives: a blend of essential oils mixed with dry herbs (PHYTO) and encapsulated sodium butyrate (BUT). This study examined how these substances affect intestinal microbiota modulation, diarrhoea incidence and persistence and performance in light weaned piglets during the nursery phase.

## Materials and methods

This study was carried out in strict accordance with the recommendations of the Guide for the Care and Use of Laboratory Animals of the National Animal Experimentation Control Board (CONCEA) in Brazil. This trial was approved by the Ethics Committee of Animal Experiments of Akei Animal Research (protocol number: 014.18).

### Description of the experiment

Two hundred weaned piglets (Camborough X PIC 337), 100 barrows and 100 females, with an average age of 20 days and an initial weight of 4.69 ± 0.56 kg, were used. The animals were housed in 40 pens in an environmentally controlled barn. They were divided into five blocks based on their initial body weight (BW), with four treatments and ten replicates per treatment. Within each block, the pigs were then allocated to pens for a balanced BW distribution. Each slatted floor pen was 2.55 m$^2$, housed five piglets and was equipped with nipple drinkers and lidded hoppers with 20 cm for each piglet.

## Experimental diets

The animals were submitted to a three-phase nutritional program: preinitial I (20 to 34 days of age), preinitial II (34 to 48 days of age) and initial I (48 to 69 days of age) (Table 1). Three performance-enhancing additives were used in different combinations in each phase throughout the study (Table 2). These were Adimix® Precision, which is 30% encapsulated sodium butyrate; Apex®5, which is a phytogenic compound composed of essential oils (41% garlic oil, 6% essential oil, cinnamic aldehyde, thymol, carvacrol and eugenol); and Enraseen 80®, an enramycin, an antimicrobial growth promoter. Test diets and water were provided ad libitum throughout the trial.

## Experimental process

Piglet weight and feed consumption were measured and recorded in each phase. These values were then used to calculate the mean body weight (BW), daily weight gain

Table 1. Ingredients and calculated composition according to the nutritional phases.

| Ingredients | Nutritional phases | | |
|---|---|---|---|
| | Preinitial I | Preinitial II | Initial I |
| Pregelatinized corn flour | 37.593 | 38.696 | 5.000 |
| Soybean meal | 18.593 | 20.100 | 28.072 |
| Corn | 0.000 | 0.000 | 58.865 |
| Fish meal 55% | 6.000 | 4.000 | 0.000 |
| Milk powder | 18.000 | 14.000 | 0.000 |
| Whey powder | 6.000 | 0.000 | 0.000 |
| Soybean oil | 0.000 | 2.386 | 2.909 |
| Dicalcium phosphate | 0.351 | 0.643 | 1.400 |
| Limestone | 0.343 | 0.000 | 0.741 |
| L-lysine | 0.755 | 0.674 | 0.567 |
| L-threonine | 0.414 | 0.211 | 0.246 |
| DL-methionine | 0.331 | 0.311 | 0.265 |
| L-valine | 0.363 | 0.194 | 0.012 |
| L-tryptophan | 0.091 | 0.082 | 0.036 |
| Salt | 0.165 | 0.441 | 0.533 |
| Vitamin Premix[1] | 0.150 | 0.150 | 0.150 |
| Mineral premix[2] | 0.100 | 0.100 | 0.100 |
| Nutrients | | | |
| Crude protein, % | 19.500 | 18.000 | 19.000 |
| Calcium, % | 0.827 | 0.584 | 0.750 |
| Available phosphorus, % | 0.520 | 0.442 | 0.350 |
| Metabolizable energy, kcal/kg | 3450 | 3450 | 3300 |
| Digestible lysine, % | 1.520 | 1.330 | 1.290 |
| Digestible methionine + cysteine, % | 0.860 | 0.778 | 0.779 |
| Digestible threonine, % | 0.958 | 0.720 | 0.840 |
| Digestible tryptophan, % | 0.258 | 0.240 | 0.230 |
| Digestible valine, % | 1.049 | 0.828 | 0.780 |

[1] levels per kg of Vitamin Premix product: vitamin A (min) 6.000 IU; vitamin D3 (min) 1.500 IU; vitamin E (min) 15.000 mg; vitamin K3 (min) 1.500 mg; vitamin B1 (min) 1.350 mg; vitamin B2 4.000 mg; vitamin B6 2.000 mg; vitamin B12 (min) 20 mg; niacin (min) 20.000 mg; pantothenic acid (min) 9.350 mg; folic acid (min) 600 mg; biotin (min) 80 mg; selenium (min) 300 mg.

[2] levels per kg of mineral premix product: iron (min) 100 mg; copper (min) 10 mg; manganese (min) 40 g; cobalt (min) 1.000 mg; zinc (min) 100 mg; iodine (min) 1.500 mg.

**Table 2. Treatments, doses, and phases.**

|  | Preinitial I (20 to 34 days) | Preinitial II (34 to 48 days) | Initial I (48 to 69 days) |
|---|---|---|---|
| **CTR** | No inclusion | No inclusion | No inclusion |
| **ENR** | 8 ppm | 8 ppm | 8 ppm |
| **BUT** | 2000 ppm | 1500 ppm | 1000 ppm |
| **PHYTO** | 150 ppm | 150 ppm | 150 ppm |

CTR: negative control; ENR: positive control (enramycin); BUT: 30% encapsulated sodium butyrate; PHYTO: essential oils.

(DWG), daily feed intake (DFI) and feed conversion rate (FCR) for each period as well as cumulatively.

Throughout the experimental period, faecal consistency was monitored, and scores were taken individually twice a day using the methodology described by Li et al. [21]. According to the scoring system, a score of 1 represented the absence of diarrhoea, that is, faeces with a normal appearance and consistency, and scores of 2 (pasty) and 3 (liquid) indicated the presence of diarrhoea. Score 3 diarrhoea data were used to calculate the diarrhoea index (the number of days with score 3 diarrhoea/total number of test days) using the example of Xiao et al. [22].

All occurrences of mortality were monitored and assessed in each phase, and all possible causes were described. On Day 42 of the trial, one animal from each pen (10 animals per treatment, 40 animals in total) was sacrificed to identify and count the caecal bacteria on a large scale by DNA sequencing. Euthanasia was performed respecting animal welfare measures. The animals were rendered unconscious using a Petrovina IS 2000 electric stunning device with two electrodes for 3 seconds (350 V and 1.3 A), and the large neck vessels were severed for bleeding. After euthanasia, 2 g samples were collected from the caecal segment of each animal. The samples were immediately transferred to individual Eppendorf tubes and frozen at -80˚C.

## Microbiome analyses

Caecal samples from the piglets were collected, and bacterial DNA was extracted using AMPureXP beads (Beckman Couleter, Brea, CA) following thermal lysis at 96˚C for 10 min. For microbiome analysis, the reads provided from the amplicons of V3V4 regions were used in the Imunova microbiome analysis pipeline.

The readings or "reads" obtained in the sequencer were analysed on the platform QIIME2 (Quantitative Insights Into Microbial Ecology) [23, 24], following a workflow from low-quality sequence removal, filtration, chimaera removal and taxonomic classification. The sequences were classified into bacterial genera through the recognition of amplicon sequence variants (ASVs), in this case, the homology between the sequences when compared against a database. To compare the sequences, the 2019 update (SILVA 138) of the SILVA database of ribosomal sequences was used [25]. To generate the classification of bacterial communities by identification of ASVs, 46927 readings were used per sample to normalize the data and not compare samples with different numbers of readings. The samples were filtered, resulting in 39 samples analysed and 1 sample cut, 819 from the CTR group (highlighted in yellow colour on the ST1) from the total.

## Statistical analysis

Each pen with five animals was an experimental unit for the growth performance parameters (BW, DFI, AWG and FCR), whereas each individual animal was the experimental unit for

microbial diversity and diarrhoea incidence and index analyses. Growth performance data were analysed using one-way analysis of variance, and then the means were compared by applying the post hoc Tukey test using R Software, version 3.5.0. Chi-square comparisons were used to evaluate diarrhoea incidence and index considering all treatments against each other, and each of the treatments against the CTR treatment. For both tests, a P value equal to or less than 0.05 was considered significant, and a P value between 0.05 and 0.10 was considered a tendency.

The statistical comparison between alpha diversities for each group analysed was performed through the nonparametric Wilcoxon test [26], considering statistically significant results to be less than 0.05 ($p < 0.05$). The statistical analysis of beta diversity was performed by per MANOVA from the QIIME2 pipeline using 10,000 permutations. All of the figures and other statistical analyses were performed using "R". The alpha diversities were calculated by the "phyloseq" [27], "vegan" [28] and "microbiome" libraries [29]. The differences in the relative abundances of taxa between the groups analysed were estimated by the Wilcoxon test [26].

## Results

Table 3 shows the effects of the treatments on growth performance for the three phases as well as for the total period. In the first phase (20 to 34 days of age), no differences were observed for any of the parameters. However, in the second phase (34 to 48 days of age), the piglets that received BUT showed higher DWG (P = 0.023) and BW (P = 0.039) than the control group (CTR). Additionally, the groups that received some additives had better FCRs than the CTR (P = 0.001). In the last phase (48 to 69 days of age), a tendency (P = 0.054) was verified; piglets

**Table 3. The effects of feed supplementation with enramycin (ENR), encapsulated sodium butyrate (BUT) and dry herbs plus essential oils (PHYT) on the growth performance of lightly weaned piglets in the nursery phase (n = 50/treatment).**

| Parameters | Treatments | | | | CV (%) | P value |
|---|---|---|---|---|---|---|
| | CTR | ENR | BUT | PHYTO | | |
| Preinitial I (20 to 34 d) | | | | | | |
| Initial weight, kg | 4.671 | 4.695 | 4.695 | 4.692 | 3.20 | 0.984 |
| DFI, kg | 0.189 | 0.189 | 0.191 | 0.193 | 10.37 | 0.956 |
| DWG, kg | 0.099 | 0.097 | 0.104 | 0.102 | 11.48 | 0.633 |
| FCR | 1.912 | 1.954 | 1.841 | 1.902 | 10.57 | 0.683 |
| Final weight, kg | 6.055 | 6.079 | 6.141 | 6.126 | 3.47 | 0.786 |
| Preinitial II (34 to 48 d) | | | | | | |
| DFI, kg | 0.364 | 0.368 | 0.375 | 0.373 | 8.35 | 0.850 |
| DWG, kg | 0.188b | 0.207ab | 0.217a | 0.209ab | 10.16 | 0.023 |
| FCR | 1.955b | 1.777a | 1.723a | 1.791a | 6.74 | 0.001 |
| Final weight, kg | 8.689b | 8.986ab | 9.188a | 9.063ab | 4.23 | 0.039 |
| Initial I (48 to 69d) | | | | | | |
| DFI, kg | 0.854 | 0.826 | 0.837 | 0.839 | 7.49 | 0.803 |
| DWG, kg | 0.506 | 0.524 | 0.519 | 0.513 | 7.51 | 0.478 |
| FCR | 1.689b | 1.572a | 1.610ab | 1.641ab | 5.72 | 0.054 |
| Final weight, kg | 19.330 | 20.021 | 20.115 | 19.842 | 5.87 | 0.448 |
| Total (20 to 69 d) | | | | | | |
| DFI, kg | 0.507 | 0.495 | 0.503 | 0.503 | 6.72 | 0.882 |
| DWG, kg | 0.299 | 0.312 | 0.314 | 0.309 | 7.51 | 0.478 |
| FCR | 1.697b | 1.584a | 1.598a | 1.632ab | 4.42 | 0.006 |

[a,b] Means with different letters on the same line are significantly different (P<0.05) or are a tendency (P >0.05 and ≤0.10) according to the Tukey test.

**Table 4. Effects of in-feed supplementation with enramycin (ENR), encapsulated sodium butyrate (BUT) and dry herbs plus essential oils (PHYTO) on the diarrhoea incidence and diarrhoea index of light weaned piglets during the nursery phase.**

| Parameters | Treatments | | | | P value |
|---|---|---|---|---|---|
| | CTR | ENR | BUT | PHYTO | |
| Number of piglets with diarrhoea score 2 | 05 | 04 | 07 | 05 | ns |
| Number of piglets with diarrhoea score 3 | 56 | 40 | 37* | 47 | ns |
| Diarrhoea index (score 3) | 1.33 | 0.95 | 0.88 | 1.11 | ns |

[a,b] Means with different letters on the same line are significantly different according to the chi-square test (P<0.05).

* Different values compared with the control group by the chi-square test (P<0.05)

treated with ENR had a higher FCR than the control and other groups. For the total period of evaluation, ENR and BUT showed better results for FCR (P<0.006) compared to the control group.

The diarrhoea incidence and index data were similar between treatments (P>0.05), but when each treatment was compared separately with the CTR, only the BUT differed statistically, being better than CTR for the diarrhea score 3 (Table 4).

In relation to the analysis of the microbiota, the phyla, classes, orders, families, genus and species with average relative abundances above 2% in at least one of the tested groups were evaluated; however, most bacteria in all taxonomic classifications did not show any differences among the specific treatments.

To generate the classification of bacterial communities by identification of ASVs, 46927 readings were used per sample. The most abundant phyla in the samples were *Firmicutes*, *Actinobacteriota* and *Bacteroidota*, as shown in Fig 1. *Firmicutes* dominated the caecal microbiota of all groups, followed by *Bacteroidetes*.

**Fig 1. Classification of the most abundant bacterial communities in the samples, through the recognition of amplicon sequence variants (ASVs).** Each color represents a different phylum.

Descriptive results of alpha and beta diversity and differential abundance of taxa are presented in Figs 2–4, respectively.

Beta diversity (Fig 3) was estimated by the parameters Bray-Curtis (p = 0.225677), Jaccard (p = 0.352565), UniFrac (p = 0.122388) and Weighted UniFrac (p = 0.489651). There was a difference in the dissimilarity of the taxa present in the samples between the ENR and BUT

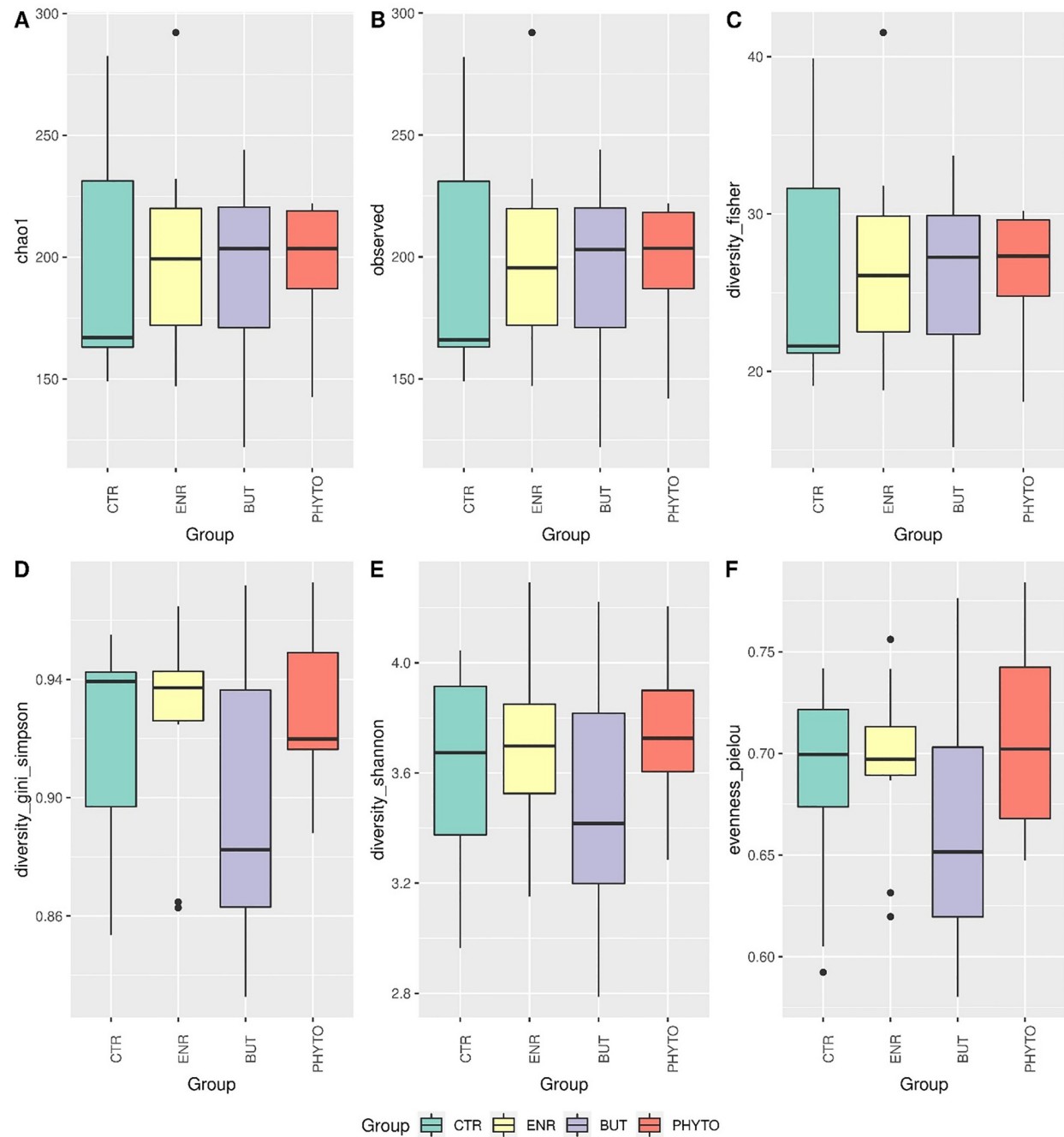

**Fig 2. Alpha-diversity estimated by the parameters Chao1 (A), observed OTUs (B), Fisher (C), Simpson index (D), Shannon entropy (E), and Evenness Pielou (F).** Statistical comparison between groups was performed using Wilcoxon's nonparametric test. Statistical results less than 0.05 were accepted as statistically significant.

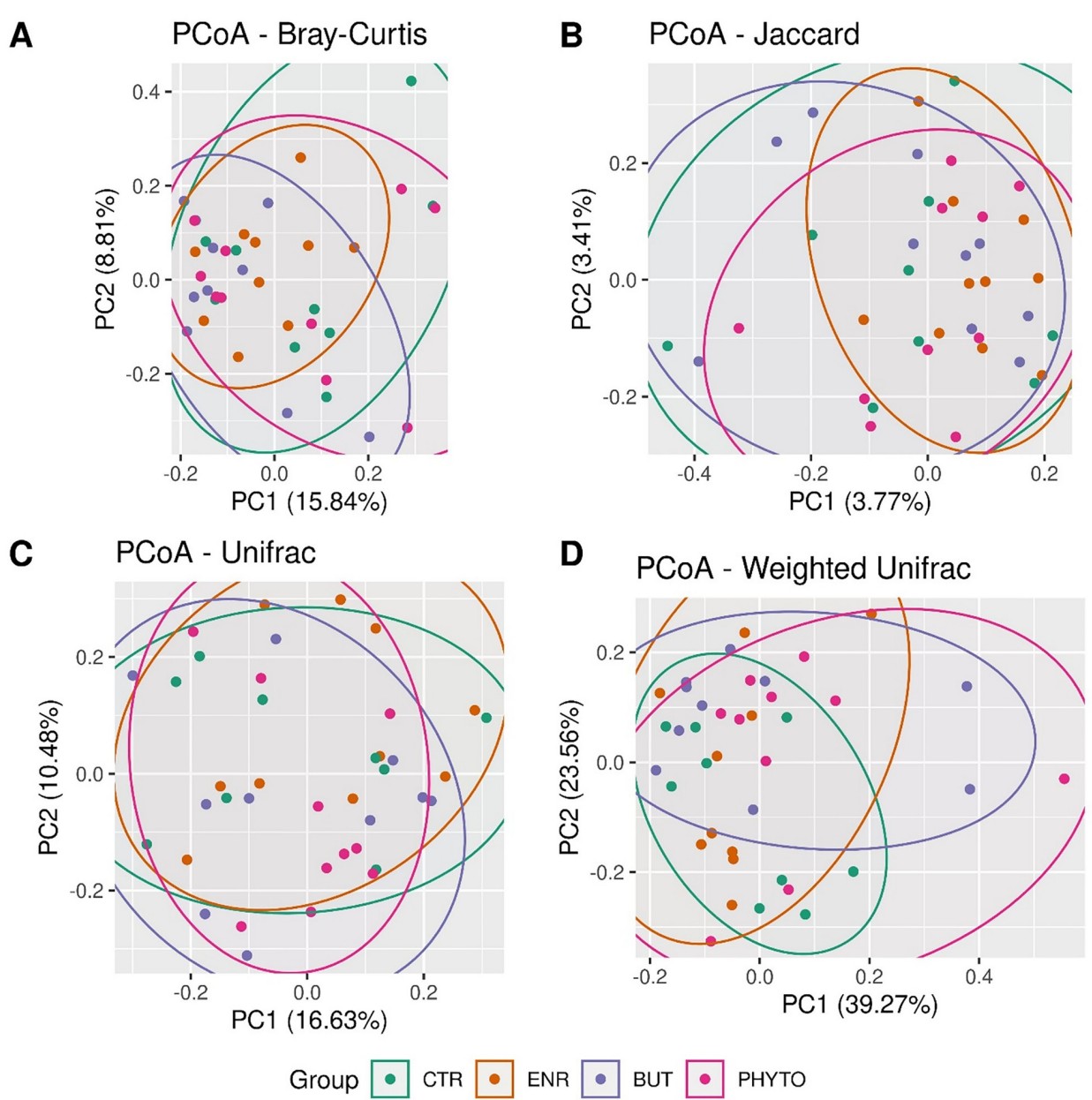

**Fig 3. Beta diversity estimated by the parameters Bray-Curtis (A), Jaccard (B), UniFrac (C), and Weighted UniFrac (D).** Coloured ellipses were automatically added through the ggforce library in R.

treatments, considering the abundance and the phylogenetic relationship between the taxa by the UniFrac parameter (Fig 3C).

Only the *Megasphaeraceae* and *Streptococcaceae* families showed statistically significant differences (Wilcoxon p<0.05) in relative abundance between the CTR and PHYTO treatments (Fig 4A) and the CTR and BUT treatments (Fig 4B), respectively.

Regarding the genus *Megasphaera*, differences were observed between the control group (CTR) and PHYTO (Fig 5A). For the *Streptococcus* genus, CTR differed from BUT (Fig 5B).

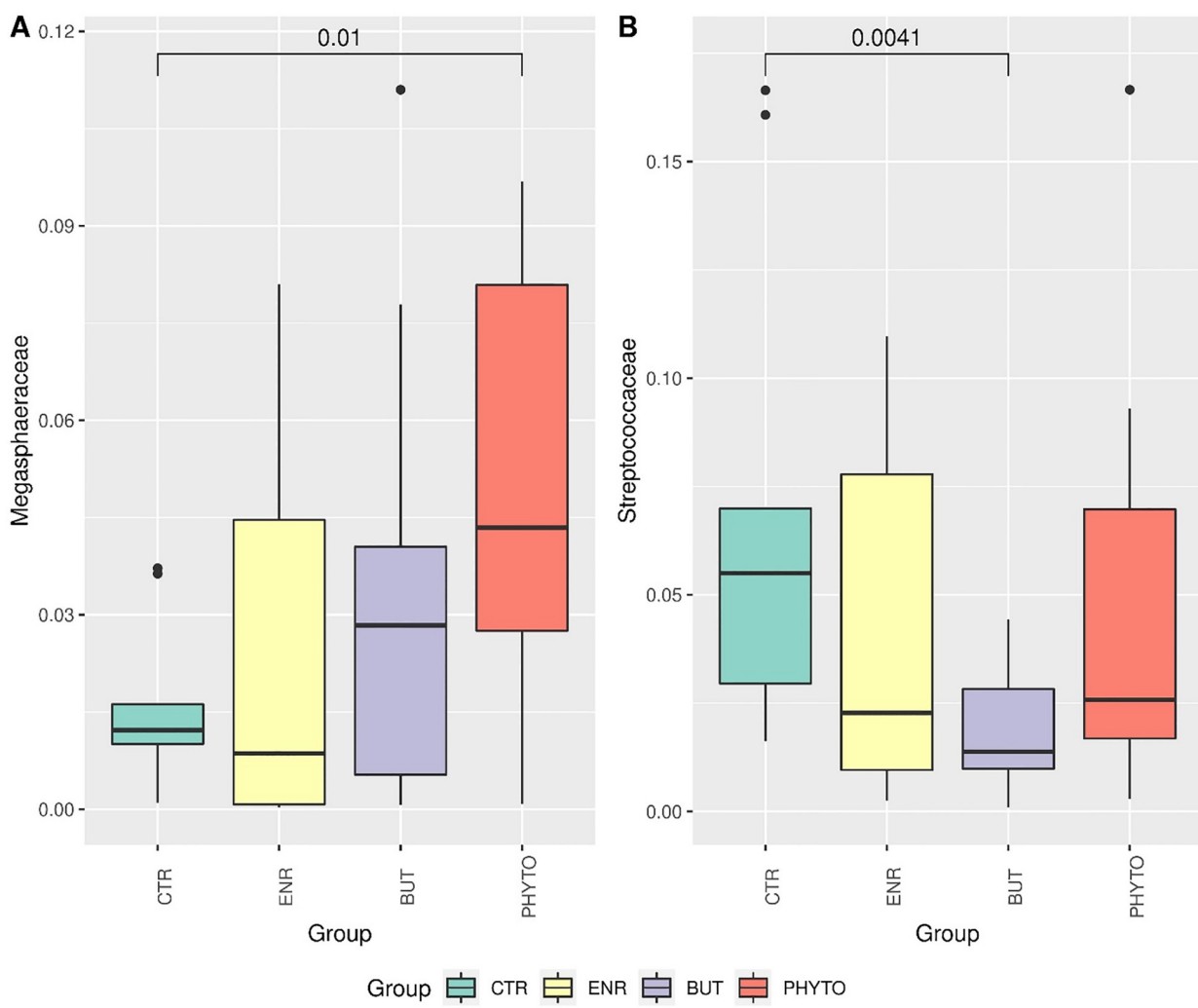

**Fig 4. Differential abundance of families *Megasphaeraceae* (A) and *Streptococcaceae* (B).** Statistical comparison between groups was performed using Wilcoxon's nonparametric test. Statistical results less than 0.05 were accepted as statistically significant.

## Discussion

During the first 14 days after weaning (preinitial I phase), there were no differences in terms of performance among the treatments, which could be the result of low voluntary feed consumption, a typical behaviour observed in many cases [30]. Each piglet ate only approximately 190 g per day, probably due to the relatively low weight of the animals, and therefore may have ingested insufficient quantities of the additives. The recovery from low feed intake observed in some postweaning conditions, such as when the piglets are light, can only occur after two weeks, when the level of energy consumed becomes similar to that recorded at preweaning, with only the piglet's maintenance needs being met [31].

Braz et al. [32] and Xu et al. [33], by contrast, found that the use of acidifiers improved DWG and FCR in the first 14 days after weaning. However, they used heavier weaned piglets, 6.69±1.89 and 8.63±1.56 kg, as opposed to the 4.69±0.56 kg used in the present study. Additionally, contrary to our results, Luise et al. [34] found that acidifiers caused improvements in feed consumption during the initial three weeks after weaning. They used formic acid at doses

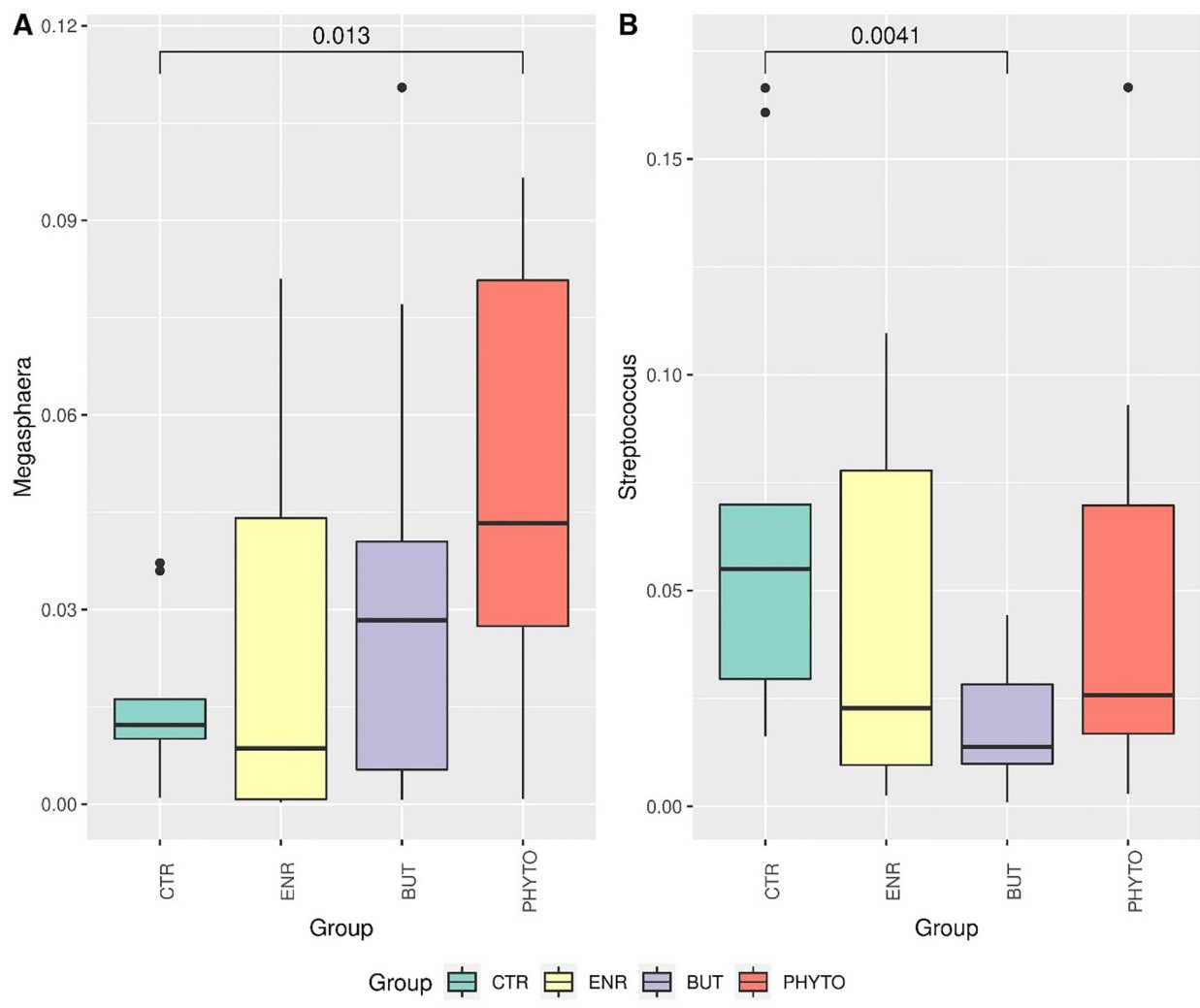

**Fig 5. Differential abundance of the genera *Megasphaera* (A) and *Streptococcus* (B).** Statistical comparison between groups was performed using Wilcoxon's nonparametric test. Statistical results less than 0.05 were accepted as statistically significant.

of 1.4 g/kg and 6.4 g/kg and compared the results to those of a basal diet. Other previous studies showed that encapsulated sodium butyrate (the acidifier used in this evaluation) had no effect (negative or positive) on this parameter for piglets in the nursery phase [32, 35].

Surprisingly, the use of PHYTO did not improve the piglets' feed consumption during this first phase either, even though it is known to improve feed palatability and digestibility and increase nutrient metabolism [36–39].

Although phytogenics may improve the performance of pigs by an average of 3% [4], the results of in vivo experiments often vary. As the products are natural, standardization is a challenge, and various factors can influence the results, such as the origin of the essential oils or the types of herbs added to the feed as well as the quantities used, not to mention the environmental conditions of the experiment [40, 41]. Likewise, the diversity of environmental and growth conditions, harvest time, maturation status, conservation duration and method, storage, synergistic and antagonistic effects and even the plant extraction method and contamination can all have an impact [42].

In the preinitial II phase, the piglets of the BUT group showed better DWG, BW and FCR than the piglets of the CTR group, which is in line with the results of a previous study by Bedford and Gong [13], which concluded that there are clear benefits to using this principle for young animals. This is due to its antimicrobial and anti-inflammatory qualities, its effects on the development and maturation of the intestinal tissue and its ability to modulate the immune response and the intestinal microbiota. Additionally, the FCR results in this phase can be related to the better use of dietary proteins [43, 44]. Sodium butyrate is converted into butyric acid after ingestion [45], and this acid can cause acidification in the intestinal lumen and increase the solubilization of some minerals that favour the development of bacterial producers of phytase (*Escherichia coli*) [43]. In turn, this enzyme has better activity in acidic environments, improving the digestion of some nutrients, including proteins. The improvement is also due to the encapsulation of sodium butyrate, which allows it to reach the last part of the GIT before its release [46].

In the initial phase I, even without the inclusion of butyrate, the BUT group still presented good results, showing better DWG and FCR compared with the CTR group, and it was similar for all parameters compared with the ENR and PHYTO groups. Our results are in accordance with those obtained by Long et al. [47], in which the use of sorbic acid plus butyrate at a dose of 2 g per kg of feed proved to be an effective substitute for antibiotic growth promoters (AGP) and generated excellent performance results (similar to those of the AGP treatment– 10 mg/kg of zinc bacitracin feed, 5 mg/kg of colistin sulphate feed and 5 mg/kg of olaquindox ration). In addition, the authors observed an improvement in intestinal morphology, namely, a reduction in the amount of *Escherichia coli* in the stool, which supports greater immune responses and reduces diarrhoea incidence.

Our results for each phase are consistent with what has been found in some previous studies; that is, encapsulated sodium butyrate has no effect on DFI during the nursery phase [32]. Therefore, the better DWG, FCR and BW observed in the preinitial II phase for the BUT treatment were reflected in the FCR for the subsequent phase and in the overall nursery period. As mentioned, the effects on the improvement of FCR can be attributed to enzyme stimulation and secretion, which improves digestion [48].

Regardless of the treatments, overall, the FCR improved during the evolution of the nursery phase. This finding is controversial; however, considering the precocious weaning age (20 d) and the light weight at weaning, the piglets probably had a hard nutritional challenge during the first weeks and presented growth compensation after this period, improving their performance, including the FCR parameter [49].

Regarding treatment with PHYTO, considering the total period of the trial, there was no difference for any parameter. Although the use of plant extracts or essential oils usually have a certain degree of positive results, the details about the commercial formula used and their photochemical and sensorial qualities are not very clear, making it difficult fully interpret the results [50, 51].

Studies on weaned piglets where the effects of plant extracts are compared to those of antibiotics require robust evaluation because in general, the benefits and efficacy of antibiotics are beyond question. Nevertheless, there is often a numerical improvement in FCR when the results are compared to the control group [40, 52]. The benefits of the phytogenic described above can be attributed to several different components present in these products that act as digestive stimulants (e.g., cinnamaldehyde, eugenol, thymol and carvacrol) [53]. These PHYTO were also taken into account in the present study. Additionally, Zou et al. [54] found that pigs treated with PHYTO showed significantly lower serum endotoxin levels, higher villi and an increase in occludine and zonula ocludens-1 in the jejunum. These results indicate an

improvement in the integrity of the intestinal barrier, which improves digestion and absorption, thus promoting performance.

Overall, the similarity of the performance results verified between the tested additives and enramycin, confirm their effectiveness against this growth promoter, although, it cannot be attributed that these same results will be observed in relation to other antibiotics as growth promoter (AGP). However, considering that the first concerns regarding the banning of AGP were linked to the increase in resistance in bacteria of human and animal origin, particularly in relation to resistance in gram negative bacteria (*Salmonella* spp. and *Escherichia coli*) [55], the AGP that are still found on the market have mostly a spectrum of action against gram positive bacteria, such as enramycin. Thus, according to our findings, and considering the largest share of AGP of this class in the market, it can be hypothesized that the PHYTO and BUT additives have the potential to determine results equivalent to this AGP class.

The diarrhoea score 3 results, exclusively considering a comparison between the CRT and BUT treatments, confirmed our expectations that BUT shows antimicrobial and anti-inflammatory activities, improves intestinal tissue development and maturation, modulates the immune response and intestinal microbiota [13], and stimulates the immune system [54]. According to Lange et al. [56] and Huang et al. [14], BUT can control pathogenic bacteria, such as *Escherichia coli* and *Salmonella spp*. but has little effect on *Lactobacillus spp*. and *bifidobacteria*. These findings are very important considering the class of the piglets evaluated. In general, light pigs at weaning presented poorly developed digestive systems [7] and are more susceptible to infectious intestinal disease and energy deficiency [8].

Our findings are broadly in line with Huang et al. [14] and Long et al. [47] when comparing the effects of using antibiotics to those of acidifiers as growth promoters for weaned piglets. Those authors also observed improvements in intestinal morphology, reductions in the amount of *Escherichia coli* in the stool, greater immune responses, and lower incidence of diarrhoea. However, considering the results of diarrhoea between CTR versus BUT treatments, we observed a reduction rate from approximately 35% in the overall phase, a better result than the previous study.

One of the main reasons for using phytogenics is its antimicrobial activity, which has a positive impact on bacterial modulation with a decrease in *Escherichia coli* count as well as diarrhoea incidents. This was shown in a study by Yan et al. [57], who used a mix of herbal extracts and observed better digestibility and immune response and a lower concentration of *Escherichia coli* in the faeces.

However, our results are different from the findings of Jiang et al. [58], who used thymol and cinnamaldehyde (the PHYTO compounds used in our study), observed a reduction in coliform and *Escherichia coli* counts, in addition to improved morphological characteristics of the small intestine, which helped preserve the intestinal barrier, leading to fewer and less intense diarrhoea incidents [54].

The results of performance and diarrhea showed by PHYTO and BUT in our study, equivalent to those observed with enramycin and identified with several studies, have an important differential, their actions were obtained with light and young weaned piglets. Lighter weaned pigs have an imature digestive system and low feed intake [59, 60], which determines a more limited intestinal mucosal cell turnover [61], compared to heavier weaned piglets, making them more vulnerable to post-weaning adversities, such as diarrhea, and more limited to performing well, even under a more energetic diet and with a higher supply of amino acids [62, 63]. This scenario supports that the action verified by the evaluated additives was positive in the face of a condition portrayed as more challenging, thus valuing their potential.

Concerning the alpha diversity of the caecal microbiota (Fig 2), the treatments did not promote significant changes in terms of richness and uniformity in the different treatments.

These findings were different from Huang et al. [14], who observed an increase in bacterial diversity (Simpson index) in the colonic lumen, with a reduction in Enterobacteriaceae (mainly Shigella) in both ileal colonic lumens.

However, for the beta diversity, there was a significant difference in the number of taxa present in the samples of the BUT and ENR groups (Fig 3). The use of encapsulated sodium butyrate (BUT) as a diet additive, which is a short-chain fatty acid that is part of the metabolic products produced by several beneficial bacterial species, has multiple benefits, highlighting the improvement of animal health and performance and feed digestibility [64].

Similarly, plant extracts or essential oils extracted from plants, among their mechanisms of action in the animal, according to Branco et al. [65], are the stimulation of digestion, changes in the intestinal microbiota, increases in digestibility and absorption of nutrients, and immunomodulatory and antimicrobial effects, the latter being mainly associated with changes in the integrity of the bacterial cell membrane and the chemical structure of the active compounds [66].

Without compromising the global stability of the microbiota, the different treatments promoted modulations in specific populations and possibly their functions in the intestinal microbiota. At the genus level, significant differences were observed regarding the relative abundance of *Megasphaera* (Fig 4A) and *Streptococcus* (Fig 4B). These influences led to significant changes in the families of these genera, as shown in Fig 5. The relative abundance of the *Megasphaera* genus increased significantly in the PHYTO treatment in relation to the negative control treatment and had the highest average observed among all groups. This genus is composed of Gram-negative cocci that are nonmotile, do not form endospores and are strictly anaerobic. They are acetic-, propionic-, butyric- and valeric acid-producing bacteria [67].

Li et al. [68] also observed an increase in the relative abundance of this genus in piglets that received a diet supplemented with phytognic compared to piglets in the CRT group, which received only the basal diet. These results correlated the application of phytogenic in the diet with an increase in body weight gain, in addition to a decrease in the incidence of diarrhoea. These considerations support the performance and diarrhoea results observed in the current evaluation.

*Streptococcus*, in turn, decreased in all groups that received treatments; however, this decrease was significant only for the BUT group. The only species of this genus identified in the samples were *S. caballi*, *S. hyointestinalis_A* and *S. suis_W* (ST9), with great emphasis on *S. hyointestinalis_A*, which was quantified in most samples. Bacteria belonging to this genus are Gram-positive, facultative anaerobic and nonspore forming. In addition, the diversity of this taxon includes commensal gut bacteria and important swine pathogens because streptococcal infections can take various forms, including meningoencephalitis, arthritis, cervical lymphadenitis, endocarditis, pneumonia and septicaemia [69].

As observed in the present study, Bernad-Roche et al. [70] also noted a significant decrease in the relative abundance of this genus in the intestinal microbiota of growing and finishing piglets that received food supplementation with sodium butyrate. In summary, we concluded that although supplementation does not change the overall richness of the microbiota composition, it may have increased the specific taxa associated with better gut health parameters.

## Conclusion

Phytogenics and encapsulated sodium butyrate are suitable for replacing enramycin in the diets of lightly weaned pigs in the nursery phase. They are just as effective as this antibiotic in improving performance, controlling diarrhoea, modulating specific caecal microbiota taxa and supporting the health of piglets.

## Supporting information

**S1 File.**
(DOCX)

**S1 Data.**
(XLS)

**S2 Data.**
(XLSX)

## Author Contributions

**Conceptualization:** David Vanni Jacob, Alexandre José Ulbrich, Tim Goossens.

**Data curation:** Caio Abércio da Silva.

**Formal analysis:** Marco Aurélio Callegari.

**Funding acquisition:** Alexandre José Ulbrich.

**Investigation:** Caio Abércio da Silva, Cleandro Pazinato Dias.

**Project administration:** Kelly Lais de Souza, David Vanni Jacob.

**Supervision:** Marco Aurélio Callegari, Gabrieli de Souza Romano, Kelly Lais de Souza, Tim Goossens.

**Writing – original draft:** Caio Abércio da Silva, Gabrieli de Souza Romano.

**Writing – review & editing:** Caio Abércio da Silva.

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
