## [Decision Letter · Decision Letter 0]

26 Oct 2022

PONE-D-22-27316Phytogenics and encapsulated sodium butyrate can replace antibiotics as growth promoters for lightly weaned pigletsPLOS ONE

Dear Dr. Caio Abércio da Silva,

Thank you for submitting your manuscript to PLOS ONE. After careful consideration, we feel that it has merit but does not fully meet PLOS ONE’s publication criteria as it currently stands. Therefore, we invite you to submit a revised version of the manuscript that addresses the points raised during the review process.

Please submit your revised manuscript by December 10, 2022. If you will need more time than this to complete your revisions, please reply to this message or contact the journal office at plosone@plos.org. Please include the following items when submitting your revised manuscript:A rebuttal letter that responds to each point raised by the academic editor and reviewer(s). You should upload this letter as a separate file labeled 'Response to Reviewers'.A marked-up copy of your manuscript that highlights changes made to the original version. You should upload this as a separate file labeled 'Revised Manuscript with Track Changes'.An unmarked version of your revised paper without tracked changes. You should upload this as a separate file labeled 'Manuscript'.

We look forward to receiving your revised manuscript.

Kind regards,

Saeed El-Ashram

Academic Editor

PLOS ONE

“The funders had no role in study design, data collection and analysis, decision to publish, or preparation of the manuscript”

Reviewers' comments:

Reviewer's Responses to Questions

**Comments to the Author**

1. Is the manuscript technically sound, and do the data support the conclusions?

Reviewer #1: Yes

Reviewer #2: Partly

2. Has the statistical analysis been performed appropriately and rigorously? 

Reviewer #1: Yes

Reviewer #2: Yes

3. Have the authors made all data underlying the findings in their manuscript fully available?

Reviewer #1: Yes

Reviewer #2: Yes

4. Is the manuscript presented in an intelligible fashion and written in standard English?

Reviewer #1: Yes

Reviewer #2: Yes

5. Review Comments to the Author

Reviewer #1: It is a complete scientific work whose primary focus is the comparison of two different approaches for feed performance and conversion. Comparison with other growth promoters could be considered for future projects due to the potential of this experimental model. I have no further suggestions for this work.

Reviewer #2: The work adequately describes the use of phytogenics and encapsulated sodium butyrate in the microbiome analysis of lightly weaned piglets. However, other response variables such as intestinal morphology, lesions, and body weight gain have yet to be analyzed to consider that phytogenics and encapsulated sodium butyrate can replace antibiotics as growth promoters. It is also necessary to indicate more strongly the new knowledge resulting from the work and how it differs from similar works.

6. PLOS authors have the option to publish the peer review history of their article (what does this mean?). If published, this will include your full peer review and any attached files.

Reviewer #1: No

Reviewer #2: **Yes: **Victor Manuel Petrone-Garcia

<quillbot-extension-portal></quillbot-extension-portal>

---

## [Author Response · Author response to Decision Letter 0]

10 Nov 2022

Dear Reviewers,

According your requests, follow the adequations that we done.

- In accordance with the reviewer's requirements, we have included two texts in lines 337 to 347 and 374 to 383 that address, respectively, the points that required clarification on the effectiveness of the evaluated additives, compared to enramycin, in terms of its effect on performance and diarrhea control; and the demonstration of what is the differential of the work, that is, what makes it different in relation to other studies that have the same characteristic.

- For the first demand, we focused on the role of tested additives against growth-promoting antibiotics with a spectrum of action against gram-positive bacteria.

- To support these two demands, we included six bibliographies, which were standardized according to the journal's rules.

---

## [Decision Letter · Decision Letter 1]

2 Dec 2022

Phytogenics and encapsulated sodium butyrate can replace antibiotics as growth promoters for lightly weaned piglets

PONE-D-22-27316R1

Dear authors,

We’re pleased to inform you that your manuscript has been judged scientifically suitable for publication and will be formally accepted for publication once it meets all outstanding technical requirements.

Kind regards,

Saeed El-Ashram

Academic Editor

PLOS ONE

Reviewers' comments:

Reviewer's Responses to Questions

**Comments to the Author**

1. If the authors have adequately addressed your comments raised in a previous round of review and you feel that this manuscript is now acceptable for publication, you may indicate that here to bypass the “Comments to the Author” section, enter your conflict of interest statement in the “Confidential to Editor” section, and submit your "Accept" recommendation.

Reviewer #1: All comments have been addressed

2. Is the manuscript technically sound, and do the data support the conclusions?

Reviewer #1: Yes

3. Has the statistical analysis been performed appropriately and rigorously? 

Reviewer #1: Yes

4. Have the authors made all data underlying the findings in their manuscript fully available?

Reviewer #1: Yes

5. Is the manuscript presented in an intelligible fashion and written in standard English?

Reviewer #1: Yes

6. Review Comments to the Author

Reviewer #1: The present investigation presents exciting information for evaluating alternatives for supplementing growth promoters through their protection and subsequent release. Consider the comparison with more growth promoters that can provide more support to this research for future experimental designs and their possible Application in the industry producing food of animal origin.

7. PLOS authors have the option to publish the peer review history of their article (what does this mean?). If published, this will include your full peer review and any attached files.

Reviewer #1: **Yes: **Inkar Castellanos-Huerta

<quillbot-extension-portal></quillbot-extension-portal>

---

## [Editor Report · Acceptance letter]

13 Dec 2022

PONE-D-22-27316R1 

Phytogenics and encapsulated sodium butyrate can replace antibiotics as growth promoters for lightly weaned piglets 

Dear Dr. da Silva:

I'm pleased to inform you that your manuscript has been deemed suitable for publication in PLOS ONE. Congratulations! Your manuscript is now with our production department. 

Kind regards, 

on behalf of

Professor Saeed El-Ashram 

Academic Editor

PLOS ONE